# Effect of a perinatal care quality improvement package on patient satisfaction: a secondary outcome analysis of a cluster-randomised controlled trial

Olivia Brunell  , Dipak Chaulagain, Ashish KC , Anna Bergström, Mats Målqvist

Department of Women's and Children's Health, Uppsala University, Uppsala, Sweden

**Correspondence to**
Dr Olivia Brunell;
olivia.brunell@kbh.uu.se

## ABSTRACT

**Objective** To investigate the effect of a quality improvement (QI) package on patient satisfaction of perinatal care.

**Design** Secondary analysis of a stepped-wedge cluster-randomised controlled trial. Participating hospitals were randomised by size into four different wedges.

**Setting** 12 secondary-level public hospitals in Nepal.

**Participants** Women who gave birth in the hospitals at a gestational age of ≥22 weeks, with fetal heart sound at admission. Adverse outcomes were excluded. One hospital was excluded due to data incompleteness and four low-volume hospitals due to large heterogeneity. The final analysis included 54 919 women.

**Intervention** Hospital management was engaged and facilitators were recruited from within hospitals. Available perinatal care was assessed in each hospital, followed by a bottle-neck analysis workshop. A 3-day training in essential newborn care was carried out for health workers involved in perinatal care, and a set of QI tools were introduced to be used in everyday practice (skill-checks, self-assessment checklists, scoreboards and weekly Plan–Do–Study–Act meetings). Refresher training after 6 months.

**Outcome measure** Women's satisfaction with care during childbirth (a prespecified secondary outcome).

**Results** The likelihood of women being overall satisfied with care during childbirth increased after the intervention (adjusted OR (aOR): 1.66, 95% CI: 1.59 to 1.73). However, the proportions of overall satisfaction were low (control 58%, intervention 62%). Women were more likely to be satisfied with education and information from health workers after intervention (aOR: 1.34, 95% CI: 1.29 to 1.40) and to have been treated with dignity and respect (aOR: 1.81, 95% CI: 1.52 to 2.16). The likelihood of having experienced abuse during the hospital stay decreased (aOR: 0.42, 95% CI: 0.34 to 0.51) and of being satisfied with the level of privacy increased (aOR: 1.14, 95% CI: 1.09 to 1.18).

**Conclusions** Improvements in patient satisfaction were indicated after the introduction of a QI-package on perinatal care. We recommend further studies on which aspects of care are most important to improve women's satisfaction of perinatal care in hospitals in Nepal.

**Trial registration number** ISRCTN30829654.

## STRENGTHS AND LIMITATIONS OF THIS STUDY

⇒ This secondary analysis focuses on patients' experience of care, an important dimension of quality of care which is often overlooked when evaluating quality improvement efforts.

⇒ The large sample size from different hospitals increases representativeness of the sample and generalisability of our results, while external validity is strengthened by the use of the WHO framework.

⇒ The stepped-wedge cluster-randomisation allowed us to eventually roll out the intervention at all sites, while mimicking a randomised controlled trial.

⇒ The exclusion of one high-volume hospital and the four low-volume hospitals deviates from the original stepped-wedged design and should be considered when interpreting the results.

⇒ We measured the WHO domains of experience of care and no other factors that are also known to affect satisfaction, such as waiting times, overcrowding and physical resources in the health facility.

## INTRODUCTION

In 2017 an estimated 2.5 million children died in the neonatal period whereof about 1 million newborn died during their first day of life.[1] Though an increasing number of women are giving birth in health facilities it has not ensured improved healthcare outcomes.[2] The quality of perinatal care provided in health facilities needs improvement to end preventable mortality.[2–4] Both Quality of Care (QoC) and approaches to its assessment are complex.[5] The definition of QoC by the WHO is 'The extent to which healthcare services provided to individuals and patient populations improve desired health outcomes' and to achieve this, healthcare must be safe, effective, timely, efficient, equitable and people-centred. Thus, healthcare must not only provide evidence-based clinical care, but by being people-centred, the care must also be provided considering

individuals preferences, needs and values and the culture of their communities.[6] This is important as patients have the right to be treated with dignity and respect, but also because people-centred care is associated with improved healthcare utilisation and health outcomes.[7] In 2016, WHO developed a framework and standards with accompanying quality measures for the quality of maternal and newborn healthcare. In this framework, QoC is divided into two interlinked dimensions: (1) *Provision of clinical care* by health workers and (2) patients' *Experience of care*.[8] *Experience of care* is a people-centred measure, a process indicator which reflects the interactions that patients have with the healthcare system.[9] It is composed of three major domains: effective communication, respect and dignity and emotional support.[6] Patient satisfaction, together with other people-centred outcomes, is an outcome of *Experience of care*.[9–11]

In 2013 the research group conducted a study of the implementation of a quality improvement (QI) package for neonatal resuscitation in a tertiary hospital in Nepal. In this study the Helping Babies Breathe (HBB) programme, developed by The American Academy of Pediatrics to improve health workers' performance on basic resuscitation,[12 13] was implemented. In addition to HBB training, the intervention included QI components, aiming to continuously reinforce the HBB protocol throughout the intervention period and to motivate both hospital management and individual health workers to be engaged in improvement.

The study, Helping Babies Breath-Quality Improvement Cycle (HBB-QIC), showed improvement in health workers adherence to neonatal resuscitation protocols and a large decrease in intrapartum stillbirths and first-day neonatal mortality.[14] These positive results are more pronounced than in previous studies of implementation of the HBB programme[15 16] and can be due to the addition of the QI components.

However, there were no significant change of overall in-hospital perinatal mortality, which indicates that efforts are also needed to improve the continued postnatal care.[17] Based on the findings from HBB-QIC, a scale-up was developed by the research group in collaboration with Ministry of Health and Population in Nepal. This is called Nepal Perinatal Quality Improvement Project (NePeriQIP) and aims to improve quality of perinatal care through increased clinical skills and knowledge in essential newborn care (ENC) and establish a structure for continuous QI.[18] The primary objective of the NePeriQIP trial was to evaluate impact on intrapartum mortality (intrapartum stillbirth and first-day mortality), and the results showed a significant reduction in intrapartum-related deaths (intrapartum stillbirths and first-day mortality) in the intervention period (adjusted OR (aOR): 0.79, 95% CI: 0.69 to 0.92).[19] In the present manuscript, we aim to evaluate the effect of NePeriQIP on the prespecified secondary outcome *patient satisfaction*.

## METHODS
### Trial design

A stepped-wedge cluster-randomised design was used,[18] where 12 hospitals were randomised by size into four wedges. Each wedge included one high-volume (>8000 deliveries a year), one medium-volume (>3000 deliveries a year) and one low-volume (>1000 deliveries a year) hospital. Each cluster had a control and an intervention period. During the first 3 months of the study period no intervention activities took place at any of the hospitals (baseline period). NePeriQIP was then introduced to one wedge with three hospitals at a time, with a 3 months delay between wedges, and eventually all hospitals were exposed to the intervention (figure 1).

### Setting

The selected 12 hospitals filled the criteria of having >1000 deliveries per year and being a referral centre for maternal and newborn care. Although most of the hospitals were located in the flatlands, they were diverse in relation to ethnicity, language and religion. All hospitals provided vaginal and caesarean deliveries, and had access to neonatal resuscitation services at birth. Skilled birth

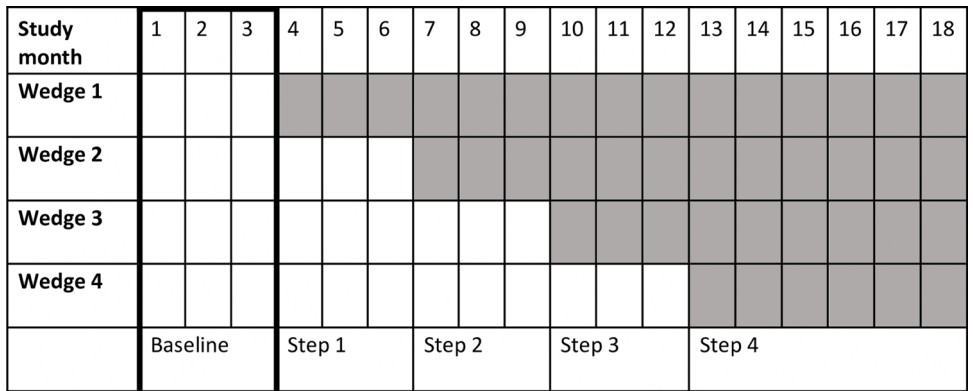

**Figure 1** Stepped-wedge design and timeline of the NePeriQIP trial. The period within bold lines represents the baseline period. The shaded period constitutes the intervention period. Each wedge contains three hospitals (clusters).

attendants (obstetricians, medical doctors, senior nurses or senior nurse midwives) led the labour units. Medical doctors led the paediatric units where sick newborns were managed in the low-volume hospitals. Paediatricians led the specialised sick newborn care units in the medium-volume and high-volume hospitals. The intrapartum-related mortality rate (intrapartum stillbirth and first day neonatal mortality) ranged from 9 to 31 per 1000 births at the hospitals (mean 13/1000 births), during the baseline period.[19]

## Participants

All pregnant women who gave birth at any of the 12 hospitals after the baseline period, who gave their oral and written consent, who were at gestational age of 22 weeks or more and who had fetal heart sound at admission were eligible for inclusion. Women who experienced stillbirths, neonatal deaths, malformations and complications during childbirth were excluded as these events could bias patient satisfaction.

## The intervention

The development of the QI package was guided by the integrated-Promoting Action on Research Implementation in Health Services (iPARIHS) framework.[20] Ministry of Health, together with the research team, developed an implementation guideline for the implementation process. In each hospital, the QI package was introduced in the following steps: First, the hospital management was oriented on the QI-package. They appointed in-hospital facilitators from among paediatricians, medical officers and nurses to facilitate the implementation, supported by the study team. The number of facilitators depended on the size of the hospital: two from low-volume, three from medium-volume and four from high-volume-hospitals. The facilitators participated in a 7-day training on the QI package, provided by the study team. Following the training, the QI facilitators assessed service readiness and availability in their respective hospital, using a checklist developed by the team, and a bottleneck analysis workshop was organised based on that. After these initial steps, a 3-day on-site training for all health staff involved in perinatal care was organised by the facilitators and the study team. The training consisted of education and skills training in ENC and introduction of a set of QI tools. The ENC component included resuscitation and immediate newborn care (thorough drying, skin-to-skin contact, delayed clamping of the cord and early initiation of breast feeding) and were taught using the HBB training manual version one.[12] In addition, lectures were given on breast feeding, infection prevention and kangaroo mother care for low birth-weight newborns. The QI tools included daily bag-and-mask ventilation skill check on an HBB mannequin (the low-cost simulator Laerdal NeoNatalie), checklist for preparations before each birth, self-assessment checklist of performed resuscitation and use of score boards comprising major indicators on neonatal resuscitation. In addition to these tools a structured model for QI, the Plan–Do–Study–Act (PDSA) cycle, was used aiming to identify and act on local problems.[21] After the training, the QI facilitators initiated weekly PDSA meetings as part of routine work at all units, involving health workers related to perinatal care. The duration of each meeting was about half an hour. A 1-day refresher training in HBB was held for health workers after 6 months.

## Conceptual framework

The WHO conceptual framework on QoC[6] and the WHO document 'Standards for improving quality of maternal and newborn care in health facilities'[8] were used to guide data collection and analyses for this manuscript. In the framework, QoC consists of the two interlinked dimensions 'Provision of care' and 'Experience of care'. Experience of care contains three domains: *effective communication, preservation of respect and dignity* and *emotional support.* Though NePeriQIP focus on clinical provision of care, a successful implementation where health workers are more engaged in the quality of perinatal care could generate better experiences among patients and manifest in patient satisfaction.

## Data collection and variables

The study period followed the Nepali calendar and was initiated on June 2017 and ran for 15 months up until October 2018. Data for this manuscript were collected through structured interviews with all included mothers on discharge, throughout the study period. An independent data collection team was established at each hospital and a structured questionnaire, developed by the study team and piloted on beforehand in a hospital later not included in the study, was used to collect the data. After completion of the data collection forms, they were sent weekly to the central research office in Kathmandu, where they were entered into a Census and Survey Processing System database by a team of independent data entry officers.

The collected data from exit interviews were not accessible for the local teams during the intervention. Results will be shared with the local teams by the published paper.

In this manuscript, we report the findings from the trial on patient satisfaction (a prespecified secondary outcome). This was measured through *Overall satisfaction* and the three dimensions of *Experience of care* (effective communication, respect and dignity and emotional support) in the WHO Quality of Care framework for maternal and newborn health.[6]

**Overall satisfaction** was measured as an aggregate score of two questions, and to be considered satisfied with services received, mothers needed to respond favourably to both questions.

1. *Overall, how satisfied are you with the services?*
   (Recorded on a Likert scale of 1 to 5. 1=very dissatisfied, 2=dissatisfied, 3=neither, 4=satisfied and 5=very satisfied. Scores were dichotomized, with an answer of 4 or more being set as satisfied).

2. *Would you recommend a friend to give birth at this hospital?* (recorded yes/ no)

**Experience of care** was investigated through a set of questions suggested as quality measures of experience of care in the WHO document 'Standards for improving quality of maternal and newborn care in health facilities'.[8]

1. Patient communication

    *Were you adequately informed by the healthcare worker about examinations, actions and decisions taken for your care throughout the hospital stay?*

    (recorded yes/no)

    *Are you satisfied with health education and information received from healthcare providers?*

    (Recorded on a Likert scale of 1 to 5. 1=very dissatisfied, 2=dissatisfied, 3=neither, 4=satisfied and 5=very satisfied. Scores were dichotomised, with an answer of 4 or more being set as satisfied).

2. Respect and dignity

    *Are you satisfied with the degree of privacy during your stay in labour and childbirth areas?*

    (Recorded on a Likert scale of 1 to 5. 1=very dissatisfied, 2=dissatisfied, 3=neither, 4=satisfied and 5=very satisfied. Scores were dichotomised, with an answer of 4 or more being set as satisfied).

    *Were you or your newborn physically, verbally or sexually abused during labour or childbirth or after birth?*

    (recorded yes/no)

    *Were you treated with respect and was your dignity preserved during your stay at the hospital?*

    (recorded yes/no)

3. Emotional support

    *Did you have a companion of your choice during labour and childbirth?*

    (recorded yes/no)

### Statistical methods

There was no a priori estimation of sample size for this secondary outcome analysis, it was calculated for the primary outcome of the NePeriQIP trial, intrapartum mortality.[19] Given that intrapartum mortality is a rare outcome, this rendered a large sample size, which allowed for the detection of small differences in the outcome of interest for the present manuscript. Pearson's $\chi^2$ test was used to analyse background characteristics of participants in the control and intervention groups, performed with Statistical Package for Social Sciences (SPSS) V.25.0. Change in satisfaction and experiences of care between the control and intervention groups were analysed by generalised linear mixed models (GLMM) using the software R (V.3.4.0). Analyses were performed with R package lme4. Adjustments were made for structural factors (cast/ethnicity and educational level) as these could be ground for discrimination and altered treatment. Intra-cluster correlations coefficients (ICCs) for each outcome were calculated. Initial analyses showed high ICC for all outcome variables, making the results inconclusive. Further analysis of the data by hospital displayed that the low-volume hospitals showed large heterogeneity. As a

consequence, the final GLMM analysis was performed on the medium-volume and high-volume hospitals. Analysis of change before and after intervention at each hospital was performed with binary logistic regression of satisfaction and experience of care (adjusted for caste/ethnicity and education level) with SPSS V.25.0. Significance level of 95% was used.

### Patient and public involvement

Patients were not directly involved in developing the trial; however, it was planned and conducted in collaboration with Ministry of Health in Nepal.

## RESULTS

A total number of 65 895 women were eligible for inclusion, whereof 946 women were excluded based on exclusion criteria. Due to data incompleteness from one of the hospitals, another 5366 women were excluded, leaving 59 583 women available for analysis. Initial analysis revealed that low-volume hospitals showed large heterogeneity and differed from the medium-volume and high-volume hospitals. As a consequence of this finding, the final GLMM analysis was performed on the medium-volume and high-volume hospitals, where 92% of deliveries took place, n=54 919 (figure 2).

### Characteristics

A majority of the women had an education higher than primary (69%) and belonged to an advantageous cast (67%). About half of the women were first-time mothers (51%) but few were adolescents (7.6%) and the coverage of antenatal care was high (80% had four antenatal care visits before giving birth). The caesarean section rate was 19% and most of them (73%) were emergency caesarean sections. The proportion of preterm births were 13% and babies with low birth weight were 14% (table 1).

### Outcomes

Women were more likely to be satisfied with care after the intervention than before (aOR: 1.66, 95% CI: 1.59 to 1.73, ICC: 0.275). The results varied between hospitals (aOR: 0.39–3.57) (online supplemental table s1) and the overall proportion of satisfaction was low. Prior to the intervention, a proportion of 58% of the women were satisfied, which increased to 62% after (table 2).

The domains of experience of care were affected in a positive direction by the intervention. Women were more likely to be *satisfied with education and information* from health workers after implementation (aOR: 1.34, 95% CI: 1.29 to 1.40, ICC: 0.167) as well as more likely to state that they had been *treated with preserved dignity and respect* (aOR: 1.81, 95% CI: 1.52 to 2.16, ICC: 0.063). The likelihood of *experience abuse during the hospital stay* decreased after the implementation (aOR: 0.42, 95% CI: 0.34 to 0.51,

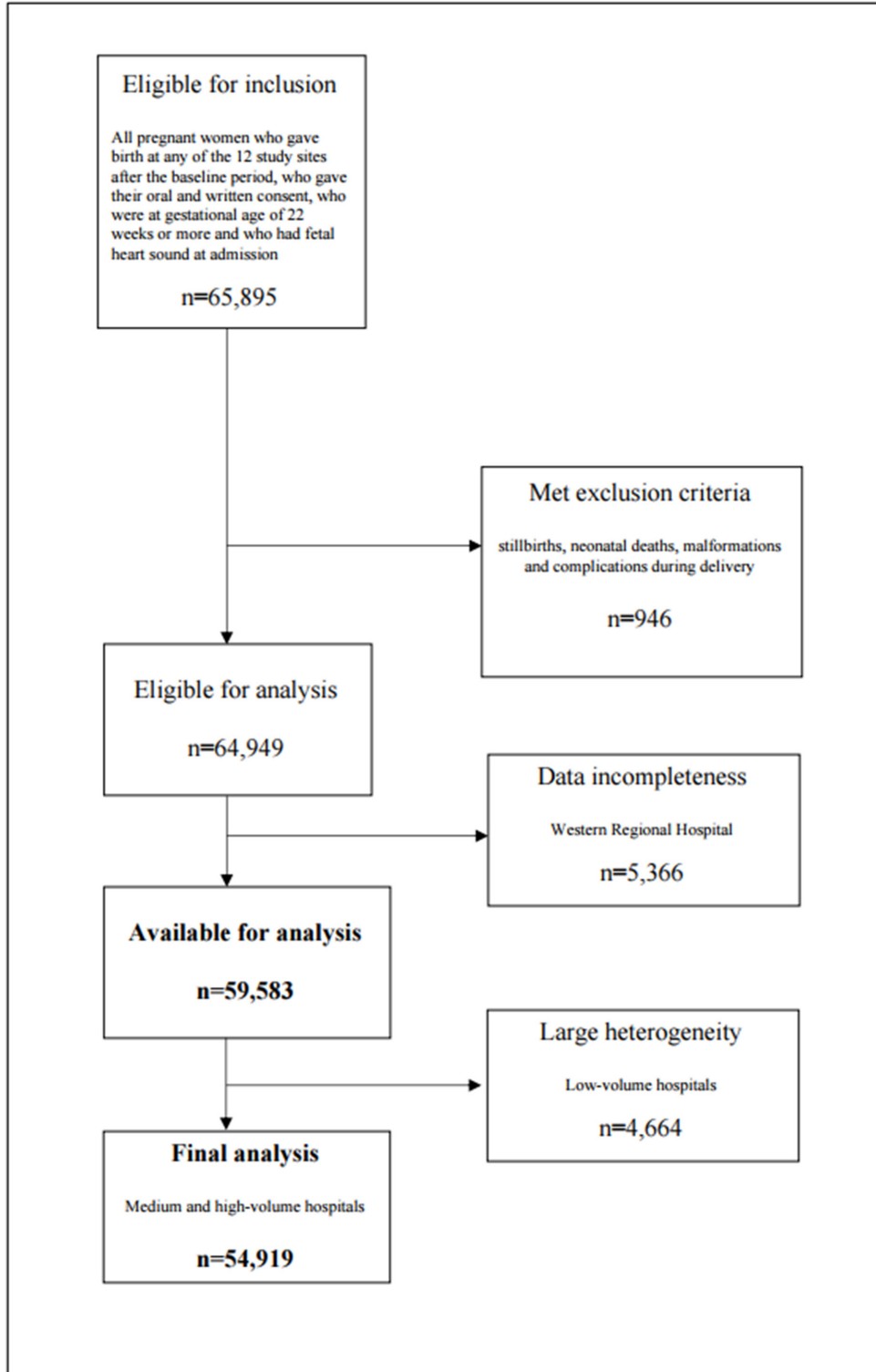

**Figure 2** Participants flow diagram.

ICC: 0.082) and the likelihood of being *satisfied with the level of privacy* increased (aOR: 1.14, 95% CI: 1.09 to 1.18, ICC: 0.136) (table 2). Two measures seemed to have been affected in a negative direction after the intervention: the likelihood of being *adequately informed about examinations, actions and decisions taken for their care* (aOR: 0.55, 95% CI: 0.53 to 0.58, ICC: 0.507) and *having a companion of choice during labour* (aOR: 0.27, 95% CI: 0.25 to 0.29, ICC:

0.687). These two measures, however, displayed relatively high ICC in the GLMM model, rendering the adjusted ORs to be inconclusive (table 2).

A very high proportion of women stated that they were treated with dignity and respect (before intervention 98.7%, after 99.1%) and very few had experienced abuse during their hospital stay (1.1% before intervention and 0.6% after). The proportion of women who stated that

**Table 1** Characteristics of study participants in control and intervention group

| Demographic characteristics | Total, n (%) | Control, n (%) | Intervention, n (%) | P value |
|---|---|---|---|---|
| Structural determinants | | | | |
| Mother's cast/ethnicity | | | | |
| Relatively disadvantaged | 19 667 (33.0) | 8537 (35.9) | 11 130 (31.1) | <0.001 |
| Relatively advantaged | 39 916 (67.0) | 15 270 (64.1) | 24 646 (68.9) | |
| Mother's education | | | | |
| Primary or less | 18 591 (31.3) | 7572 (31.9) | 11 019 (30.9) | 0.006 |
| Secondary or higher | 40 829 (68.7) | 16 139 (68.1) | 24 690 (69.1) | |
| Intermediate determinants | | | | |
| Adolescent mother (≤19 years) | | | | |
| No | 55 061 (92.4) | 22 026 (92.5) | 33 035 (92.3) | 0.415 |
| Yes | 4522 (7.6) | 1781 (7.5) | 2741 (7.7) | |
| Elder mother (>35 years) | | | | |
| No | 58,902(98.9) | 23,557 (98.9) | 35 345 (98.8) | 0.082 |
| Yes | 681 (1.1) | 250 (1.1) | 431 (1.2) | |
| Parity | | | | |
| Nulli | 30 610 (51.4) | 12 400 (52.1) | 18 210 (50.9) | 0.005 |
| Multi | 28 973 (48.6) | 11 407 (47.9) | 17 566 (49.1) | |
| ≥Four antenatal care visits | | | | |
| No | 11 700 (19.9) | 5136 (21.9) | 6564 (18.5) | 0.000 |
| Yes | 47 131 (80.1) | 18 273 (78.1) | 28 858 (81.5) | |
| Spontaneous vaginal | | | | |
| No | 13,998 (23.3) | 5294 (22.2) | 8704 (24.3) | <0.001 |
| Yes | 45 585 (76.7) | 18 513 (77.8) | 27 072 (75.7) | |
| Elective caesarian section | | | | |
| No | 56 563 (94.9) | 22 413 (94.1) | 34 150 (95.5) | <0.001 |
| Yes | 3020 (5.1) | 1394 (5.9) | 1626 (4.5) | |
| Emergency caesarian section | | | | |
| No | 51 559 (86.5) | 20 846 (87.6) | 30 713 (85.8) | <0.001 |
| Yes | 8024 (13.5) | 2961 (12.4) | 5063 (14.2) | |
| Preterm | | | | |
| No | 51 579 (86.6) | 21 740 (91.3) | 29 839 (83.4) | <0.001 |
| Yes | 8004 (13.4) | 2067 (8.7) | 5937 (16.6) | |
| Low birth weight | | | | |
| No | 46 997 (86.2) | 20 540 (87.4) | 26 457 (85.3) | <0.001 |
| Yes | 7524 (13.8) | 2949 (12.6) | 4575 (14.7) | |
| Sex of baby | | | | |
| Boy | 32 132 (54.1) | 12 771 (53.9) | 19 361 (54.2) | 0.391 |
| Girl | 27 288 (45.9) | 10 940 (46.1) | 16 348 (45.8) | |

$\chi^2$ test to detect group differences before and after NePeriQIP intervention start. $n_{tot}$=59 583.

they were adequately informed about their care were 76.6% before intervention and 80.7% after. However, not many women had a companion of choice during labour (16.9% before and 19.0% after), and the proportions of women who were satisfied with the information and education from health workers, and with the level of privacy, were also low (43.3% before and 49.3% after, 51.6% before and 51.0% after, respectively) (table 2).

**Table 2** Binary logistic regression for full sample and GLMM analysis correcting for clustering (by hospital) displaying adjusted ORs

| | Control, n (%) | Intervention, n (%) | P value | GLMM aOR (95% CI)* | ICC |
|---|---|---|---|---|---|
| **Patient satisfaction** | | | | | |
| **Are you satisfied with services and would you recommend a friend?** | | | | | |
| No | 9070 (42.1) | 12 694 (38.0) | | Ref | |
| Yes | 12 485 (57.9) | 20 670 (62.0) | 0.00 | 1.66 (1.59–1.73) | 0.275 |
| **Patient experience** | | | | | |
| **Were you adequately informed by the care provider about examinations, actions and decisions taken for your care?** | | | | | |
| No | 5021 (23.4) | 6437 (19.3) | | Ref | |
| Yes | 16 439 (76.6) | 26 882 (80.7) | 0.00 | 0.55 (0.53–0.58) | 0.507 |
| **Are you satisfied with education/ information?** | | | | | |
| No | 12 227 (56.7) | 16 922 (50.7) | | Ref | |
| Yes | 9328 (43.3) | 16 422 (49.3) | 0.00 | 1.34 (1.29–1.40) | 0.167 |
| **Are you satisfied with level of privacy?** | | | | | |
| No | 10 433 (48.4) | 16 350 (49.0) | | Ref | |
| Yes | 11 122 (51.6) | 17 014 (51.0) | 0.167 | 1.14 (1.09–1.18) | 0.136 |
| **Were you or your newborn physically, verbally or sexually abused during labour or childbirth or after birth?** | | | | | |
| No | 21 308 (98.9) | 33 149 (99.4) | | Ref | |
| Yes | 247 (1.1) | 215 (0.6) | 0.00 | 0.42 (0.34–0.51) | 0.082 |
| **Were you treated with respect and was your dignity preserved during your stay at the hospital?** | | | | | |
| No | 275 (1.3) | 290 (0.9) | | Ref | |
| Yes | 21 280 (98.7) | 33 074 (99.1) | 0.00 | 1.81 (1.52–2.16) | 0.063 |
| **Did you have a companion of your choice during labour and childbirth?** | | | | | |
| No | 17 837 (83.1) | 26 991 (81.0) | | Ref | |
| Yes | 3623 (16.9) | 6328 (19.0) | 0.00 | 0.27 (0.25–0.29) | 0.687 |

$\chi^2$ test for group differences, n=54 919.
*GLMM models adjusted for clustering (by hospital) and caste/ethnicity and education level.
GLMM, Generalised Linear Mixed Method; ICC, intraclass correlation.

## DISCUSSION

The quality of perinatal care in health facilities needs to be improved to save newborn lives.[2–4] Measuring patient satisfaction is important in several aspects to improve QoC. It can reveal gaps in the health system to address, it creates accountability among healthcare providers and stakeholders, it is useful to guide and evaluate QI efforts, and in addition satisfaction is known to affects patients care-seeking behaviour and compliance.[9 22–24] Our analysis shows that after the introduction of the QI package of NePeriQIP, the likelihood that women were satisfied with perinatal care increased. Satisfaction is an outcome measure of experience of care,[9] and in our analysis the people-centred quality measures of experience of care suggested by WHO were affected in a positive direction, along with satisfaction. The exact mechanisms behind our results are not clear as the intervention did not specifically focus on people-centred skills. However, besides that NePeriQIP focused on improvement of provision of care through training, it also included QI tools to strengthen professional attitudes among health workers and sustain change in practice. Self-evaluation, insight in the ongoing

progress and involvement in problem-solving promote a higher level of learning which can create a deeper understanding among health workers and make them pay more attention to what they are doing and why.[25] This can improve their capacity to inform and educate patients, to handle stressful situations and provide more time and engagement to attend to interpersonal interactions with patients. With increased knowledge and continuous discussions concerning care of the mothers and newborn, healthcare workers might also get more sensitised and empathic towards their patients. A recent implementation study conducted in India, also adopting the PDSA approach and facilitation to improve quality of care, could show improvements in satisfaction among women who had recently given birth similar to our results.[26]

Satisfaction is a complex phenomenon influenced by several determinants, and we cannot conclude that the improvement in satisfaction is explained solely by improvement in the measures of experience of care that we have studied. Other known determinants of satisfaction are structural elements (ie, cleanliness of the health facility and availability of human recourses, medicines and supplies) and outcome-related determinants such as health status of the mother and newborn.[27] Nevertheless, it is previously shown that the interpersonal components of experience of care dominate satisfaction with maternity care in Nepal and other low-income countries.[27 28]

Though we could see an increased satisfaction after the intervention, the proportion of satisfied women both before and after was low. This despite that women tend to report more satisfied with care if interviewed early after childbirth, biased by social desirability and the joy of having a healthy baby[29] and previous studies of satisfaction in Nepal and India has shown higher levels.[26 30 31] We do not know why the women were not more satisfied with care but some factors can be considered. First, we sought to reduce the influence of social desirability by our definition of satisfaction (overall satisfaction combined with whether to recommend the facility to a friend). In addition, other determinants that was not the scope of our analysis, could be of importance. Other studies of satisfaction with perinatal care in Nepal have shown that having a protected waiting area and having an opportunity to ask questions are positively associated with patient satisfaction, while overcrowding and long waiting times has been shown to increase the likelihood of dissatisfaction.[28 31]

We adjusted for the structural determinants cast/ethnicity and educational level, which can be grounds for discrimination, altered treatment and outcomes. The intermediate determinants displayed in table 1 were not considered grounds for discrimination and thus not included in the further analysis. Given the complexity of the stepped-wedge design, no further multilevel analyses were performed

The main strengths of our secondary outcome analysis are the large sample and the stepped-wedge cluster randomisation. The latter allowed us to eventually roll out the intervention at all sites, while mimicking a randomised-controlled trial. It was also guided by WHO framework which strengthens external validity.

There are some limitations worth mentioning. We excluded women with adverse outcomes since this could over shadow the experience of care and skew our results. We had to exclude one high-volume hospital due to data incompleteness and the four low-volume hospitals due to initial analysis showing high ICC. This deviates from the original stepped-wedged design and should be considered when interpreting the results. We only focused on measuring the WHO domains of experience of care and no other factors that are also known to affect satisfaction, such as waiting times, overcrowding, patients' expectations, provision of medical care or physical resources in the health facility.[28 32] A more in-depth understanding of the women's satisfaction could have been achieved by adding qualitative interviews.

## CONCLUSION

Measuring patient satisfaction is an important aspect of quality-of-care development and evaluation. We found that after the introduction of a QI package (NePeriQIP), women giving birth in health facilities were significantly more likely to be satisfied with care. Given the large sample size and the modest increase in the proportion of overall stated satisfaction, the results should be interpreted with caution. As satisfaction with perinatal care in Nepal still needs improvement, we recommend further studies on aspects of care that are of most importance to improve patient satisfaction.

**Acknowledgements** The authors would like to thank the management and health care staff of all participating hospitals as well as facilitators, mentors, data collectors and the data management team at Golden Community. Thanks to Abhishek Gurung, Elisha Joshi, Sunil Gajurel, Prajwal Poudel and Asmita Paudel for facilitating the QI implementation process in hospitals. Special thanks to Omkar Basnet for excellent work in data management and to Regina Gurung for participating in managing the project. Lastly, we would like to express our gratitude towards all participating mothers.

**Contributors** OB, MM, AB and AKC conceptualised the study. AKC and DC supervised data collection and management. OB and MM analysed data and prepared the first draft. All authors, reviewed, commented and approved the final version of the manuscript. MM is the guarantor.

**Funding** This work was supported by Swedish Research Council (2016-05621), the Laerdal Foundation for Acute Medicine, Norway (40198) and Einhorn Family Foundation, Sweden. The funders had no role in study design, data collection and analysis, decision to publish or preparation of the manuscript.

**Competing interests** None declared.

**Patient and public involvement** Patients and/or the public were not involved in the design, or conduct, or reporting or dissemination plans of this research.

**Patient consent for publication** Not applicable.

**Ethics approval** This study involves human participants and ethical approval was obtained from the Nepal Health Research Council (ref 26-2017). The data used for this secondary outcome analysis were collected through face-to-face interviews after obtaining written informed consent from the participants. The identity of the participants was coded after data collection, and the code key was stored in a safe environment that only the project management had access to.

**Provenance and peer review** Not commissioned; externally peer reviewed.

**Data availability statement** Data are available upon reasonable request. Deidentified participant data used in the analysis of this manuscript will be available upon request. The study protocol has been published.

**ORCID iDs**
Olivia Brunell http://orcid.org/0000-0003-0343-5221
Ashish KC http://orcid.org/0000-0002-0541-4486
Mats Målqvist http://orcid.org/0000-0002-8184-3530

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
