## [Reviewer comments · BMJ Open]

ARTICLE DETAILS

TITLE (PROVISIONAL)	Effect of a perinatal care quality improvement package on patient satisfaction: a secondary outcome analysis of a cluster randomized control trial
AUTHORS	Brunell, Olivia; Chaulagain, Dipak; KC, Ashish; Bergström, Anna; Målvist, Mats

VERSION 1 – REVIEW

REVIEWER	Das, Manoja The INCLEN Trust International
REVIEW RETURNED	21-Nov-2021

GENERAL COMMENTS	The authors have presented the results of QI effort for perinatal care on patient satisfaction in Nepal. The manuscript needs attention as per the following observations/suggestions. 1. Methods: 1.1. PDSA cycles: How many PDSA cycles were implemented and the duration of these cycles should be specified. 1.2. Data collection and variables: Was the data collected accessible for the local team? How the data results were shared with the local teams? 1.3. Statistical methods: sample size calculation. A post-hoc power calculation should be presented to indicate the ability of the sample for detection of minimum change. Any subgroup analysis done for the mothers of newborns required SNCU admission vs those who did not? Any subgroup analysis done for the mothers who underwent LSCS delivery vs vaginal delivery? Any cutoff for the satisfaction level considered for the analysis? 2. Results 2.1. 10 of the 12 hospitals are reflected in Table S1. Were two hospitals excluded? The limitation mentions one hospital excluded. 2.2. Any data on the sustenance of the satisfaction level across the months would be useful. 3. Discussion May include the references from similar QI studies conducted in India. 1. Das MK, Arora NK, Dalpath SK, Kumar S, Kumar AP, Khanna A, et al. (2021) Improving quality of care for pregnancy, perinatal and newborn care at district and sub-district public health facilities in three districts of Haryana, India: An Implementation study. PLoS ONE 16(7): e0254781. https://doi.org/10.1371/journal.pone.0254781 2. Agarwal R, Chawla D, Sharma M for the QI Haryana Study Collaboration, et al Improving quality of care during childbirth in
--

	primary health centres: a stepped-wedge cluster-randomized trial in IndiaBMJ Global Health 2018;3:e000907.
--	--

REVIEWER	Bogren, Malin University of Gothenburg
REVIEW RETURNED	01-Feb-2022

GENERAL COMMENTS	Dear Authors, Thank you very much for the opportunity of reviewing your manuscript on the effect of Nepal Perinatal Quality Improvement Project on patient satisfaction: a cluster randomized control trial. It is an interesting study presenting data about women's satisfaction and experience of services after a quality improvement intervention at several of hospitals in Nepal. It is an important paper and I would like to congratulate the authors of a well performed work of this paper. I have a few comments to be considered before being published. P4 line 14-17: You write that "Both the definition of Quality of Care (QoC) and approaches to its assessment are subjects for discussion and there are no universal consensus on these matters" and uses a reference from 1966. While in the following sentence you refer to the WHO definition of Quality of Care 2015 and its quality measures. This seems as contradicting information. A suggestion is to either remove the sentence in lines 14-17, or use a more updated reference which presents evidence on that there is no universal consensus on these matters. P4 line 51: is reference 12 the correct reference to Helping Babies Breathe (HBB) program, developed by The American Academy of Pediatrics? P5 line 60: who are these skilled birth attendants? (Nurse-Midwives, Nurses, Physicians, Auxiliary nurse-midwives) P6 line 59: you may want to explain what a HBB mannequin is, (example the low-cost simulator, NeoNatalie) P7 line 7: who are these health staff? The same as the skilled birth attendants or different personnel? P7 line 24: here you use the term health workers, who are they? If these are all the same health personnel, it is suggested that you use the same term across the paper. P7 line 36: Did you consider using an already validated tool for measuring women's satisfaction and experiences? If not, please explain why? P8 line 28: here is the term health care providers being used, see above comments In General: please consider using the term childbirth or/and birth, instead of delivery. Women give birth, not deliver.
---

REVIEWER	Kelly, Dervla University of Limerick, Graduate Entry Medical School
REVIEW RETURNED	25-Feb-2022

GENERAL COMMENTS	This is an interesting topic. I recognize that the researchers are limited by the data collected. However, I have some concerns about the impact of the differences between the intervention and control group on the ability of the study to detect meaningful differences between groups. Methods Page 7: "piloted on before" on who. Please complete sentence
--

	Adjustments were made for structural factors (cast/ethnicity and educational level): please explain why these are adjusted for Initial analyses showed high ICC for multiple variables, making the results inconclusive. Which variables? Did the removal of the low vol hospitals change the correlation coefficient? Further analysis of the data by hospital displayed that the low-volume hospitals showed large heterogeneity: please add summary to supplementary material. Results Table 2: Please report unadjusted Chi2-test differences for all outcomes between intervention and control Discussion Page 13: Line 52: I am concerned about the differences between the intervention and control groups in this study and do not think omitting them from the analysis as they are intermediate determinants is well justified. Further justification and consideration of their role in satisfaction is required and their impact on the results. Do the researchers believe the intervention improved the satisfaction of patients given the bias in design? Was their issues with implementation on a hospital level? Seems like more research about implementation is required. Likewise, the comment about cast/ethnicity and educational level altering care seems worth exploring further, from a QI improvement perspective to make the finding more generalizable and situated in context.
--	---

VERSION 1 – AUTHOR RESPONSE

Reviewer: 1

Dr. Manoja Das, The INCLEN Trust International

Comments to the Author:

The authors have presented the results of QI effort for perinatal care on patient satisfaction in Nepal. The manuscript needs attention as per the following observations/suggestions.

Methods:

PDSA cycles: How many PDSA cycles were implemented and the duration of these cycles should be specified.

- Response: We have now clarified in the manuscript that PDSA meetings were introduced as part of the routine work, that the frequency of PDSA meetings was once every week and the duration of each PDSA meeting was about half an hour.

Data collection and variables: Was the data collected accessible for the local team? How the data results were shared with the local teams?

- Response: Data processing and analysis was an important part of the intervention (i.e the score boards comprising major indicators on neonatal resuscitation) and was used in PDSA-cycles. Collected data from exit interviews however, were not shared during intervention. The analyzed data (data results) are shared with the local teams by sharing the published papers/ reports. Clarified in the manuscript.

Statistical methods:

sample size calculation. A post-hoc power calculation should be presented to indicate the ability of the sample for detection of minimum change.

- Response: Given the large sample-size it can be fair to assume that very small differences can be detected. Therefore, we do not see a need for further power calculations. Clarified in manuscript.

Any subgroup analysis done for the mothers of newborns required SNCU admission vs those who did not?

Any subgroup analysis done for the mothers who underwent LSCS delivery vs vaginal delivery?

- Response: No subgroup analysis was performed in this study.

Any cutoff for the satisfaction level considered for the analysis?

- Response: Yes, there was a cutoff level. As specified in the manuscript, the overall satisfaction was measured as an aggregate score of two questions;

(1) Overall, how satisfied are you with the services?

(Recorded on a Likert scale of 1 to 5. 1= very dissatisfied, 2= dissatisfied, 3= neither, 4= satisfied, 5= very satisfied. Scores were dichotomized, with an answer of 4 or more being set as satisfied).

(2) Would you recommend a friend to deliver at this hospital?

(recorded yes/ no)

To be considered satisfied with services received, mothers needed to respond favorably to both questions.

Results

10 of the 12 hospitals are reflected in Table S1. Were two hospitals excluded? The limitation mentions one hospital excluded.

- Response: In table S1, seven hospitals are reflected. A total of 5 hospitals was excluded as described in the results section. "Due to data incompleteness from one of the hospitals, another 5,366 women were excluded, leaving 59,583 women available for analysis. Initial analysis revealed that low-volume hospitals showed large heterogeneity and differed from the medium and high-volume hospitals. As a consequence of this finding, the final GLMM analysis was performed on the medium and high-volume hospitals where 92% of deliveries took place, n= 54,919 (Figure 3)."

In table S1 we have now deleted the hospital numbers, which might have been confusing.

Any data on the sustenance of the satisfaction level across the months would be useful.

- Response: Yes, sustainability is highly interesting, but given the before and after design in this sub-study we have not analyzed the process. Process-evaluation and evaluation of sustainability of the NePeriQIP will be undertaken, but is not the scope of this sub-study.

Discussion

May include the references from similar QI studies conducted in India.

1. Das MK, Arora NK, Dalpath SK, Kumar S, Kumar AP, Khanna A, et al. (2021) Improving quality of care for pregnancy, perinatal and newborn care at district and sub-district public health facilities in three districts of Haryana, India: An Implementation study. *PLoS ONE* 16(7): e0254781.

<https://doi.org/10.1371/journal.pone.0254781>

2. Agarwal R, Chawla D, Sharma M for the QI Haryana Study Collaboration, et al. Improving quality of care during childbirth in primary health centres: a stepped-wedge cluster-randomized trial in India *BMJ Global Health* 2018;3:e000907.

- Response: Thank you for sharing these interesting articles. The first one has been added as a reference to our study. The second one has not been added, as it explores a different context (primary health centers) and different outcomes. For this reason, though it showed gaps in patients experiences, it is not considered comparable to our study.

Reviewer: 2

Dr. Malin Bogren, University of Gothenburg

Comments to the Author:

Dear Authors,

Thank you very much for the opportunity of reviewing your manuscript on the effect of Nepal Perinatal Quality Improvement Project on patient satisfaction: a cluster randomized control trial. It is an interesting study presenting data about women's satisfaction and experience of services after a quality improvement intervention at several of hospitals in Nepal. It is an important paper and I would like to congratulate the authors of a well performed work of this paper.

I have a few comments to be considered before being published.

P4 line 14-17: You write that "Both the definition of Quality of Care (QoC) and approaches to its assessment are subjects for discussion and there are no universal consensus on these matters" and uses a reference from 1966. While in the following sentence you refer to the WHO definition of Quality of Care 2015 and its quality measures. This seems as contradicting information. A suggestion is to either remove the sentence in lines 14-17, or use a more updated references which presents evidence on that there is no universal consensus on these matters.

- Response: We have considered your suggestions and removed that there is no universal consensus on these matters, as we are actually using the WHO-definition. However, we've kept that it is complex, and added a more updated reference on this.

P4 line 51: is reference 12 the correct reference to Helping Babies Breathe (HBB) program, developed by The American Academy of Pediatrics?

- Response: We have now added two other, more accurate references

P5 line 60: who are these skilled birth attendants? (Nurse-Midwives, Nurses, Physicians, Auxiliary nurse-midwives)

- Response: They are obstetricians, medical doctors, senior nurses or senior nurse midwives, now clarified in the manuscript

P6 line 59: you may want to explain what a HBB mannequin is, (example the low-cost simulator, NeoNatalie)

- Response: We have added "(the low-cost simulator Laerdal® NeoNatalie)"

P7 line 7: who are these health staff? The same as the skilled birth attendants or different personnel?

P7 line 24: here you use the term health workers, who are they? If these are all the same health personnel, it is suggested that you use the same term across the paper.

- Response: Thank you for making us aware using different terms. They are all the same and are now referred to as "health workers" across the paper.

P7 line 36: Did you considering using an already validated tool for measuring women's satisfaction and experiences? If not, please explain why?

- Response: The QI- package was developed using WHO recommendations, documents and guidelines, especially "Standards for improving quality of maternal and newborn care in health facilities". The measures we have used are suggested in this document.

P8 line 28: here is the term health care providers being used, see above comments

- Response: This was a direct translation from the data collection form in Nepali, we have now changed into health workers to be consistent.

In General: please considering using the term childbirth or/and birth, instead of delivery. Women give birth, not deliver.

- Response: The term delivery has been changed into childbirth/birth as per the suggestion, were appropriate.

Reviewer: 3

Dr. Dervla Kelly, University of Limerick

Comments to the Author:

This is an interesting topic. I recognise that the researchers are limited by the data collected.

However, I have some concerns about the impact of the differences between the intervention and control group on the ability of the study to detect meaningful differences between groups.

- Response: Thank you. Yes, we acknowledge the limitations in the dataset.

Methods

Page 7: "piloted on before" on who. Please complete sentence

- Response: The data collection tools were piloted in a hospital with maternity service in Lalitpur. The tools were piloted on the eligible women (available during the pilot) who gave consent to participate. Clarified in the manuscript.

Adjustments were made for structural factors (cast/ethnicity and educational level): please explain why these are adjusted for

- Response: We believe that discrimination is an important factor that can alter the experience of care and satisfaction. As structural determinants are ground for discrimination and differed between groups, we adjusted for them. Clarified in manuscript.

Initial analyses showed high ICC for multiple variables, making the results inconclusive. Which variables? Did the removal of the low vol hospitals change the correlation coefficient?

- Response: It showed high ICC for all outcome variables, manuscript updated. Since a high ICC makes the results inconclusive, the comparison of correlation coefficient before and after removal of hospitals is not relevant.

Further analysis of the data by hospital displayed that the low-volume hospitals showed large heterogeneity: please add summary to supplementary material.

- Response: The process has been described in Figure 3.

Results

Table 2: Please report unadjusted Chi2-test differences for all outcomes between intervention and control

- Response: P-values added in table 2.

Discussion

Page 13: Line 52: I am concerned about the differences between the intervention and control groups in this study and do not think omitting them from the analysis as they are intermediate determinants is well justified. Further justification and consideration of their role in satisfaction is required and their impact on the results.

- Response: To avoid the risk of over adjustment, we did not consider the intermediate determinants included in data collection to be grounds for discrimination and these were thus excluded from the further analysis. Clarifications have been made both to the text and through re-organisation of Table 1.

Do the researchers believe the intervention improved the satisfaction of patients given the bias in design? Was their issues with implementation on a hospital level? Seems like more research about

implementation is required.

- Response: Yes, we agree that further research on the implementation process is needed. Given the rather modest improvements in the proportion of overall satisfaction it is important to be cautious in the interpretation of results. This is now clarified in the conclusion. Further process evaluations and qualitative inquiry might reveal new insights.

Likewise, the comment about cast/ethnicity and educational level altering care seems worth exploring further, from a QI improvement perspective to make the finding more generalizable and situated in context.

- Response: We completely agree with this comment, further explorations into the role of discrimination are warranted.

VERSION 2 – REVIEW

REVIEWER	Das, Manoja The INCLEN Trust International
REVIEW RETURNED	07-May-2022

GENERAL COMMENTS	The authors have addressed the comments.
--

REVIEWER	Bogren, Malin University of Gothenburg
REVIEW RETURNED	13-Apr-2022

GENERAL COMMENTS	Dear authors, thank you for considering my comments. I am happy to recommend this paper for publication. I would however ask you to consider using the term birth/childbirth instead of delivery across the paper. For instance in the abstract, instead of delivery care, the sentence could read care during birth. In the discussion, at ref 26 the text could instead read among women who had recently given birth. Wishing you all the best with your future research.
--

REVIEWER	Kelly, Dervla University of Limerick, Graduate Entry Medical School
REVIEW RETURNED	28-Apr-2022

GENERAL COMMENTS	No further comments. Thank you
--------------------------------

VERSION 2 – AUTHOR RESPONSE

Reviewer: 2

Dr. Malin Bogren, University of Gothenburg

Comments to the Author:

Dear authors, thank you for considering my comments.

I am happy to recommend this paper for publication. I would however ask you to consider using the term birth/childbirth instead of delivery across the paper. For instance in the abstract, instead of delivery care, the sentence could read care during birth. In the discussion, at ref 26 the text could instead read among women who had recently given birth.

Wishing you all the best with your future research.

- Response: Thank you! We now use the term childbirth/birth throughout the manuscript.

Reviewer: 3

Dr. Dervla Kelly, University of Limerick

Comments to the Author:

No further comments. Thank you

Reviewer: 1

Dr. Manoja Das, The INCLEN Trust International

Comments to the Author:

The authors have addressed the comments.